# Validation of Turbulence Intensity as Simulated by the Weather Research and Forecasting Model off the U.S. Northeast Coast

Sheng-Lun Tai[1], Larry K. Berg[1], Raghavendra Krishnamurthy[1], Rob Newsom[1], and Anthony Kirincich[2]

[1]Pacific Northwest National Laboratory, Richland, WA, 99352, U.S.A.

[2]Woods Hole Oceanographic Institution, Falmouth, Massachusetts, 02543, U.S.A.

*Correspondence to*: Sheng-Lun Tai (sheng-lun.tai@pnnl.gov)

**Abstract**

Turbulence intensity (TI) is often used to quantify the strength of turbulence in wind energy applications and serves as the basis of standards in wind turbine design. Thus, accurately characterizing
the spatiotemporal variability of TI should lead to improved predictions of power production. Nevertheless, turbulence measurements over the ocean are far less prevalent than over land due to challenges in instrumental deployment, maintenance, and operation. Atmospheric models such as mesoscale (weather prediction) and large-eddy simulation (LES) models are commonly used in wind energy industry to assess the spatial variability of a given site. However, the TI derivation from
atmospheric models have not been well examined. An algorithm is proposed in this study to realize online calculation of TI in the Weather Research and Forecasting (WRF) model. Simulated TI is divided into two components depending on scale, including sub-grid (parameterized based on turbulence kinetic energy (TKE)) and grid resolved. Sensitivity of sea surface temperature (SST) on simulated TI is also tested. An assessment is performed by using observations collected during a field campaign conducted
from February to June 2020 near the Woods Hole Oceanographic Institution 's Martha's Vineyard Coastal Observatory. Results show while simulated TKE is generally smaller than lidar-observed value, wind speed bias is usually small. Overall, this leads to a slight underestimation in sub-grid scale estimated TI. Improved SST representation subsequently reduces model biases in atmospheric stability as well as wind speed and sub-grid TI near the hub height. Large TI events in conjunction with mesoscale weather systems
observed during the studied period pose a challenge to accurately estimate TI from models. Due to notable

uncertainty in accurately simulating those events, it suggests summing up sub-grid and resolved TI may not be an ideal solution. Efforts in further improving skills in simulating mesoscale flow and cloud systems are necessary as the next steps.

## 1 Introduction

While the number of wind turbines installed offshore in U.S. waters is small, it is expected to continuously escalate for the foreseeable future (U.S. Department of Energy (2021)). Therefore, it is critical to accurately simulate the wind resource, as well as the turbulence in offshore environments. Wind power generation is sensitive to atmospheric turbulence in addition to wind speed (Yang et al. 2017; Berg et al. 2019; Vanderwende and Lundquist 2012; St Martin et al. 2016; Wharton and Lundquist 2012).

Atmospheric turbulence impacts the loads on the turbine and, ultimately, the life span of the wind turbine (Mucke et al. (2011). Even in the ambient environment without any turbulence, the turbine itself generates turbulence via its wake (Wu and Porté-Agel 2012; Brand et al. 2011; Porté-Agel et al. 2020; Hansen et al. 2011).

Turbulence is generally largest near the surface where the wind shear and buoyancy are generally

largest and it is closely connected to stability in the planetary boundary layer (PBL) (Rodrigo et al. 2015; Bardal et al. 2018; Garratt 1994). Compared to conditions over land, the sea surface is relatively smooth leading to smaller amounts of shear-generated turbulence near the surface. In addition, while the continental PBL is generally unstable during the day, the diurnal variation of marine PBL is not evident but dependent on thermal gradient between sea surface and atmosphere above. Thus, the atmosphere is

generally less turbulent offshore than onshore (e.g. Bodini et al. 2020). Nevertheless, strong turbulence-producing events such as hurricanes, winter storms, and mesoscale convective systems can impact offshore wind farms.

In wind energy applications, turbulence is often quantified using the turbulence intensity (TI) (Bodini et al. 2020; Barthelmie et al. 2007), and TI is the basis of standards used in wind turbine design (e.g.,

Shaw et al. 2022). Accurately characterizing the spatiotemporal variability of TI should lead to improved predictions of power production. Earlier studies discussed how TI may influence power production of

turbines (Bardal and Sætran 2017; Kaiser et al. 2007; Saint-Drenan et al. 2020; Clifton and Wagner 2014). It is shown that power production during periods of high or low TI can vary by up to 20% (Lundquist and Clifton 2012). However, turbulence and stability measurements over the ocean are made far less

frequently than over land as it is challenging to maintain and operate instruments over the open ocean for long durations.

Atmospheric models such as weather-prediction and large-eddy simulation (LES) models can potentially bridge this gap as they can simulate turbulence based on atmospheric and surface conditions for any region over the globe. While LES models are too computationally expensive to simulate long

periods of time, mesoscale meteorological models are much more efficient and can also estimate turbulent properties such as turbulent kinetic energy (TKE) by applying an applicable turbulence parameterization. Nevertheless, the derivation of TI from atmospheric models have been rarely examined. Since multiple model uncertainties may contribute to TI bias under various conditions, comprehensive observational datasets are desired for model validation and help quantify the errors.

This study addresses these shortcomings to derive TI from a mesoscale weather model and access its performance in an offshore environment. We implement an online calculation of TI in the standard version of the Weather Research and Forecasting (WRF) model and complete a quantitative assessment of simulated TI using observations collected during a field campaign conducted from February to June 2020 near the Woods Hole Oceanographic Institution's (WHOI)'s Martha's Vineyard Coastal

Observatory (MVCO). Simulations incorporating high-resolution sea surface temperature (SST) are performed to evaluate the impact of SST on changing the atmospheric stability and simulated values of TI.

The paper is organized as follows: Observational datasets used in this study are described in Section 2. Details of model configuration, TI derivations, and experimental design are provided in the Section 3.

Section 4 covers our primary findings including (a) wind and turbulence profiles near sea surface, along with corresponding air-sea temperature difference; (b) the sensitivity of simulated wind and turbulence to SST forcing; (c) the relationship between TI and bulk Richardson number; and (d) the role of model-resolved TI. To conclude, summary and discussion are given in the Section 5.

## 2 Observational data

A U.S. Department of Energy (DOE) and Bureau of Ocean Energy Management supported field campaign was conducted from February through June 2020 near the Woods Hole Oceanographic Institution's (WHOI)'s Martha's Vineyard Coastal Observatory (MVCO, Austin et al. 2000) with the goal of evaluating the performance of DOE's lidar buoys (Gorton and Shaw 2020; Krishnamurthy et al. 2021; Sheridan et al. 2022). The MVCO is a purpose-built facility for conducting detailed atmospheric and

oceanic research. A major component of the MVCO is the Air-Sea Interaction Tower (ASIT), built near the vicinity of the Rhode Island and Massachusetts Wind Energy Areas (Figure 1). At the site, a suite of wind energy specific measurements were made, including a pair of cup anemometers at the top of the tower [26 meters above mean sea level (ASL)], a wind vane at 23 meters ASL, and a Windcube v2 vertically profiling LIDAR (hereafter reference lidar) on the main platform located 13 meters ASL. The

centers of the reference lidar range gates are: 53, 60, 80, 90, 100, 120, 140, 160, 180, and 200 meters ASL. Windcube v2 measures line-of-sight radial velocity sequentially along four cardinal directions, with a zenith angle of 28 degrees, and a fifth beam is vertically pointed. The temporal resolution along each beam direction is 1.25 Hz. A carrier-to-noise ratio of -23 dB was used to filter the raw radial velocity measurements. Two DOE buoys equipped with a Doppler lidar (also a Windcube v2), surface met station,

wave sensors and current profilers were deployed within 200 m of the ASIT tower (hereafter buoy lidar). With predominant summer winds from ~250 degrees and winter winds from ~320 degrees, the configuration of the buoy lidar and ASIT was designed to minimize wind wakes on the buoys from the tower. The reference heights of the buoy lidars matched the reference lidar heights. In addition, the air temperature at 4 meters (using a Rotronic/MP101A sensor) and sea surface temperature was collected

using a YSI thermometer on the buoy lidar (41.33°N 70.57°E and 41.32°N 70.57°W) and are used to determine the buoyancy component of local stability. In this analysis, we use the reference lidar TI calculations for comparison to simulated TI, since the buoy induces additional motion which results in higher uncertainty in TI estimates (Gottschall et al. 2017; Kelberlau et al. 2020). Additional research is ongoing to improve TI calculations using buoy lidar measurements.

TI is defined as the standard deviation of the horizontal wind speed ($\sigma_U$) divided by the average horizontal wind speed over a time interval ($\overline{U}$):

$$TI = \frac{\sigma_U}{\overline{U}}, U = \sqrt{u^2 + v^2} \tag{1}$$

Horizontal wind vectors ($u$ and $v$) from the reference lidar were calculated at ~1 Hz using the raw radial velocity measurements along the cardinal directions. Although the winds are reconstructed at 1 Hz, the winds are a combined product of a trailing ~ 4 seconds of sampled radial velocity measurements. Any

radial velocity measurement below -23 dB signal to noise ratio threshold was filtered. The reconstructed 1 Hz horizontal wind speed measurements are used to calculate TI from the Doppler lidar as shown in Eq. 1 over 10-minutes. TKE is generally estimated as

$$TKE = \frac{1}{2}(\sigma_u^2 + \sigma_v^2 + \sigma_w^2), \tag{2}$$

where $\sigma_u^2$ $\sigma_v^2$ and $\sigma_w^2$ are the variances of zonal, meridional, and vertical wind components, respectively. For 10-minute averaged TKE from reference lidar profilers, horizontal velocity variance was estimated

using the 1Hz u and v components of velocity and the vertical velocity variance was estimated using the lidar central beam staring vertically up (like Arthur et al. 2022). Only the measurements with greater than 90% data availability over the 10-minute averaging periods are used.

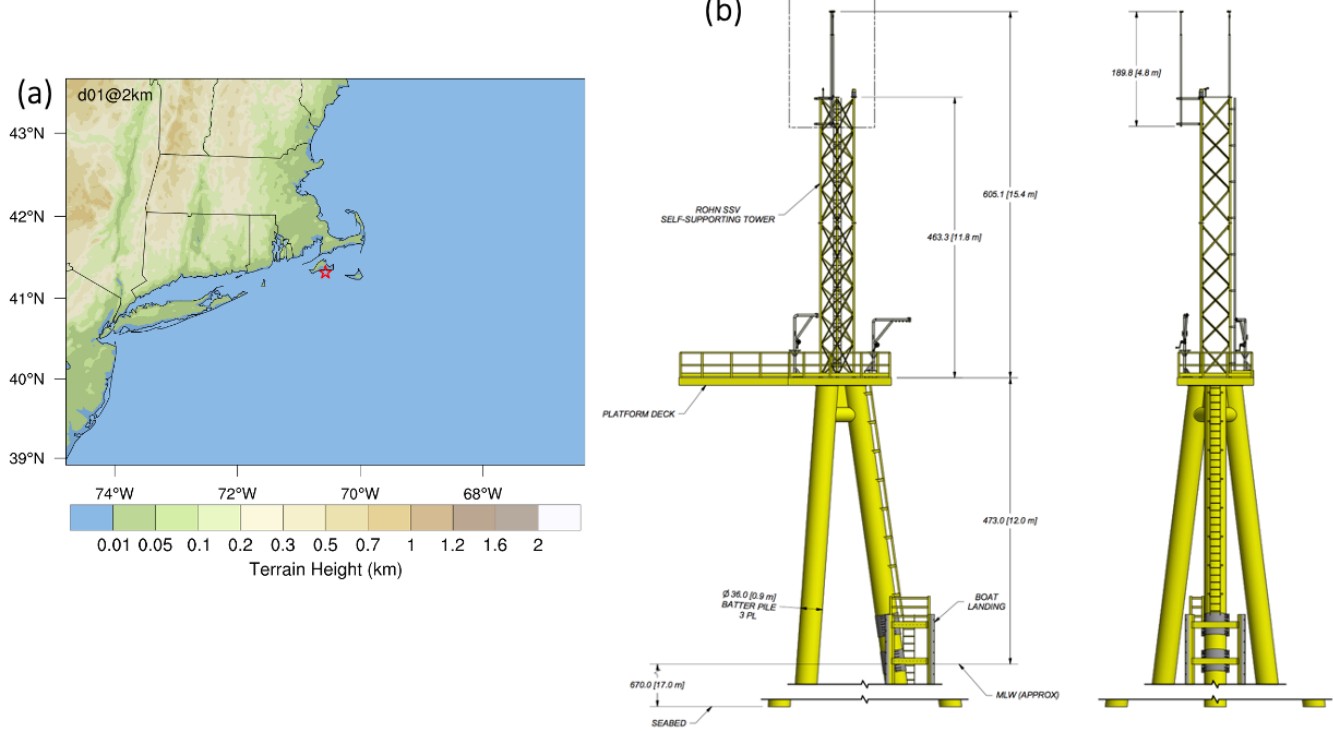

**Figure 1: (a) map depicting the WRF domain used in this study. Red star denotes the location of MVCO ASIT. Color shading represents the terrain height in kilometer. (b) schematics of WHOI's Air Sea Interaction Tower (ASIT) including all pertinent elevations and dimensions of the tower structure and fixed MetOcean sensors. The reference lidar system is located outboard on the platform deck.**

## 3 Model descriptions

### 3.1 Configuration

The model configuration, including  horizontal and vertical grid spacing, spin-up time and selections of atmospheric, SST forcing, as well as PBL and surface layer parameterizations can contribute to the uncertainty in simulated offshore wind (Bodini et al. 2021; Siedersleben et al. 2020; Hahmann et al. 2015; Chang et al. 2015; Sward et al. 2023; Optis et al. 2021). In this study, the version 4.2 WRF model is used to simulate offshore near surface winds. A single model domain centered on the MVCO site is used that encompasses part of the north-eastern U.S. and adjacent oceans (Figure 1). The choices of model physics

parameterizations for this study are consistent with the setup for the 20-year wind resource dataset released by the National Renewable Energy Laboratory (NREL) (Optis et al. 2020). Optis et al. (2020)
conducted a series of model sensitivity experiments with respect to surface layer and PBL parameterizations, reanalysis data, and SST forcing. The results of their model assessment indicated that the largest uncertainty is associated with the choice PBL parameterizations, and the Mellor-Yamada-Nakanishi Niino (MYNN) boundary layer parameterization (Nakanishi and Niino 2009) generally outperforms the Yonsei University (YSU, Hong et al. 2006) scheme off the east coast of North America.
Hence the MYNN boundary layer parameterization, as well as other parameterizations described by Optis et al. (2020) were used for generating the CA20 dataset as well as our simulations. The model uses a horizontal grid spacing of 2 km and a stretched vertical coordinate with 60 levels. There are approximately 10 model levels between the surface and 200 m. The Eta (Ferrier) microphysics parameterization, MYNN surface layer parameterization, Unified Noah land-surface parameterization (Chen and Dudhia 2001), and
the RRTMG longwave and shortwave radiation parameterization (Iacono et al. 2008) are employed. Initial and boundary conditions are taken from the NOAA's High-Resolution Rapid Refresh (HRRR; Benjamin et al. (2016)) product. The HRRR analysis has several advantages over other coarse-resolution reanalysis products. For instance, 1) the model core of HRRR, the WRF model, is identical with what we use in this study, and 2) it has a grid spacing of 3 km, which is very close to the grid spacing used here (2 km) to
match the CA20 dataset, and 3) it is constrained hourly by assimilating radar observations including Doppler velocity and reflectivity, which reduces the uncertainties in the prediction of precipitating clouds. All these advantages would primarily mitigate model uncertainties in part due to issues in model balance and spin-up.

**3.2 Diagnostics of turbulence intensity (TI)**

Realistically, observed TI occurs across a range of spatial and temporal scales. Application of the WRF model leads to an artificial separation of motions that are explicitly represented by the model and turbulence treated by the boundary-layer parameterization, thus both grid-resolved and sub-grid motions can give rise to TI. These two contributions to TI will be referred to model-resolved and sub-grid TI, respectively. A new algorithm is implemented to extract the wind variation and mean wind speed over a

10-min window allowing for the calculation of both sub-grid and model-resolved TI. Since the three-dimensional components of turbulent wind speed cannot be obtained through the boundary layer parameterization, the sub-grid TI is derived by leveraging TKE (Eq. 2) prognosed by the MYNN boundary-layer parameterization.

The derivation of TI has been proposed in several studies including a recent document of Larsén (2022). In this report, the relation between TI and TKE is derived using the Kaimal boundary-layer turbulence model (Kaimal and Finnigan 1994). However, since the TI derivation in that work divides horizontal wind into along- and cross-turbine wind components rather than zonal and meridional winds that used in WRF, we decided to apply the TI formula used in Shaw et al. (1974), Wharton and Lunquist

(2011), and Bodini et al. (2020):

$$TI = \frac{\sqrt{\sigma_u^2 + \sigma_v^2}}{\overline{U}} \tag{3}$$

Our analysis using the lidar-measured wind variances collected at the MVCO ASIT over the study period indicates $\sigma_w^2$ (variance of vertical wind) is generally much smaller than $\sigma_u^2$ and $\sigma_v^2$ (Figure 2a) regardless of stability (Figure 2b). In most of the conditions, the fraction of $\sigma_w^2$ in total wind variance ($\frac{\sigma_w^2}{\sigma_u^2 + \sigma_v^2 + \sigma_w^2}$) is no greater than 0.2 (not shown). Moreover, it is found the data points exceed 0.2 mostly occur during

neutral conditions, rather than during periods of unstable conditions that we normally would expect.

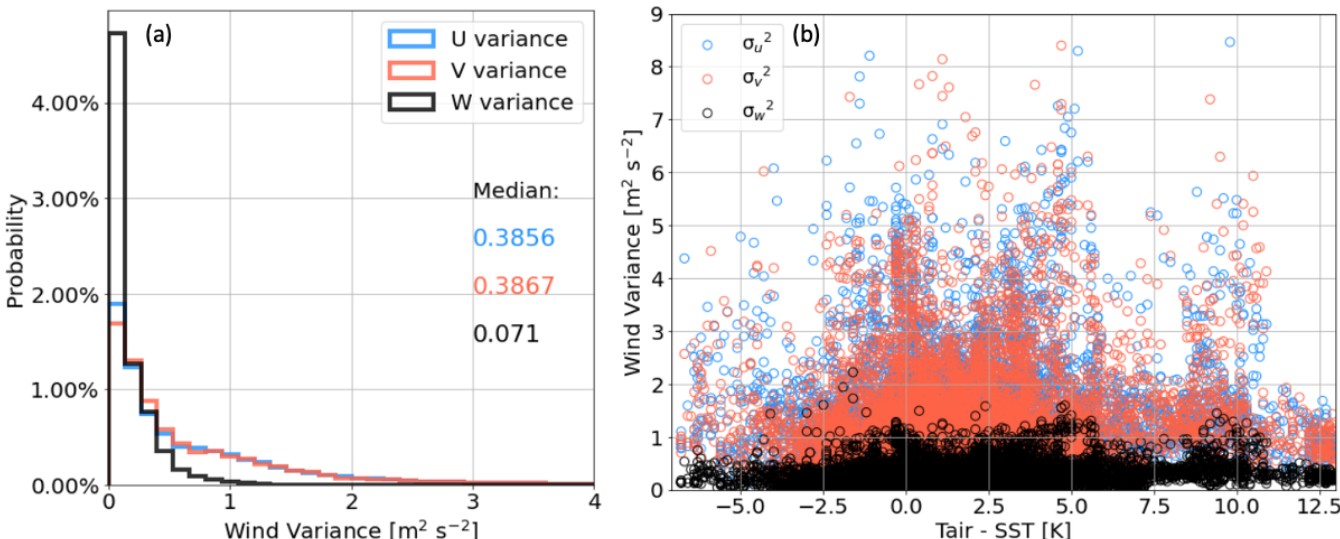

**Figure 2: The lidar-measured wind variances ($\sigma_u^2$, $\sigma_v^2$, and $\sigma_w^2$) collected at the MVCO ASIT from January to mid-June in 2020. The data is displayed in the format of (a) power density and (b) in function of air-sea temperature difference (atmospheric stability).**

Therefore, Eq. (3) can be approximated by the form of

$$TI \cong \frac{\sqrt{2 * TKE}}{\overline{U}} \qquad (4)$$

The instantaneous sub-grid TI at any model time step can then be obtained by a scaling of the mean wind speed in a 10-min window following Eq. (4). A new variable was added to store the sub-grid TI and to have values written to the WRF model output files. In addition, the model was also modified to compute the model-resolved TI by adding calculations of running means and variances of the horizontal velocity components (zonal and meridional) within 10-min window that are then used to compute and output the resolved TI. A summary of two components of TI from WRF model is given in Table 1.


**Table 1. Summary of comparison between two types of TI output from WRF model.**

| | Sub-grid TI | Model-resolved TI |
|---|---|---|
| **Dynamic scale** | Turbulence | Mesoscale wind fluctuation |
| **Solution in model** | PBL scheme (MYNN) | Resolved wind components |
| $\sqrt{\sigma_u^2 + \sigma_v^2}$ | TKE (assume $\sigma_w^2$ is negligible) | Variance of u, v in 10-min window |
| $\bar{U}$ | Mean horizontal wind speed in 10-min window | Mean horizontal wind speed in 10-min window |

## 3.3 Experimental design

To facilitate comparison against observations collected during the field campaign near the MVCO ASIT, the WRF model simulations were performed for February through June of 2020. The "baseline" experiment is a concatenation of a series of 36-hour simulations. Each of these simulations is initialized at 00 UTC and continuously integrated for 36 hours. To avoid model spin-up issues, the first 12-hours each individual simulation are discarded, and the resulting 24-hours results are retained for the analysis. Since the near-surface atmospheric stability can strongly influence turbulence intensity at hub-height, the uncertainty in representation of sea surface temperature should also be considered. Despite the HRRR analysis better represents convective-scale structures that other coarse-resolution reanalysis products as mentioned earlier in Section 3.1, it does not provide corresponding SST forcing data. Therefore, a sensitivity experiment (named "sstupdate" hereafter) is conducted to examine the variability induced by replacing the SST forcing in the model.

Note that Optis et al. (2020) used two SST products including the Operational Sea Surface Temperature and Sea Ice Analysis (OSTIA, resolution: 0.05°) data set (Donlon et al. 2012) and the National Center for Environmental Prediction (NCEP) Real-Time Global (RTG) SST product (Grumbine 2020, resolution: ~0.083°). Here, we use the NASA Jet Propulsion Laboratory (JPL) Level 4 MUR Global Foundation Sea Surface Temperature Analysis (V4.1, GHRSST; Chin et al. (2017)) which has an even higher spatial resolution (0.01°) than either the OSTIA and NCEP RTG. The SST analysis product

assimilates satellite data with a range of resolutions including the Moderate Resolution Imaging Spectroradiometer (MODIS) infra-red (1-km), AVHRR infra-red (5 – 9 km), microwave (25 km) and in-situ measurements (pointwise). The Multi-Resolution Variational Analysis (MRVA) technique is employed to reconstruct fast-moving fine-scale features as well as fill the large-scale data void. Note that since HRRR analysis data does not include SST, the baseline simulation uses the climatological SST provided with WRF as its forcing.

## 4 Results

### 4.1 Near sea-surface wind, turbulence, and air-sea temperature gradient

The simulated wind speed (WSPD), wind direction (WDIR), TI, TKE, and air-sea temperature gradient (air temperature at 2m above the surface minus sea surface temperature (SST)) are compared against data measured by tower-mounted lidar and buoy deployed at the ASIT site. We use the air-sea temperature gradient as a proxy of atmospheric stability near the sea surface. Negative values indicate unstable (convective) conditions, whereas stable conditions are likely when the temperature gradient is positive.

Here we highlight the comparisons by showing the results from March and May in 2020 as examples. The panels in Figures 3 and 4 show a composite month-long comparison of near-surface properties between "baseline" simulations and the tower-mounted lidar for March and May, respectively. Note the model profiles in the figure only extend to 200 m to match the range of the lidar. The results show the model reproduces the major "convective" (negative values of Tair–SST; red patches) events in March. While the simulated TKE is generally smaller than observed, both observations and simulations have larger TI and TKE during convective periods, indicating the model's PBL scheme reacts reasonably well in response to the varied lower boundary conditions. During stable periods (positive values of Tair–SST; blue patches), simulated TKE and TI decrease dramatically with height. This indicates there is weak turbulence and limited vertical mixing in these cases. In addition, the seasonal variability of air-sea temperature gradient is evident. For instance, the convective events in May were much shorter and weaker

than those seen in March. This is most likely due to more frequent cold air outbreak in March than in late spring. The cold-air out breaks are likely strongly convective periods, due the advection of cold air over relatively warm water. Cases with near-zero air-sea temperature gradients present a greater challenge as they have less tolerance in modelling inaccurate air-sea temperature differences and fluxes within the boundary layer.

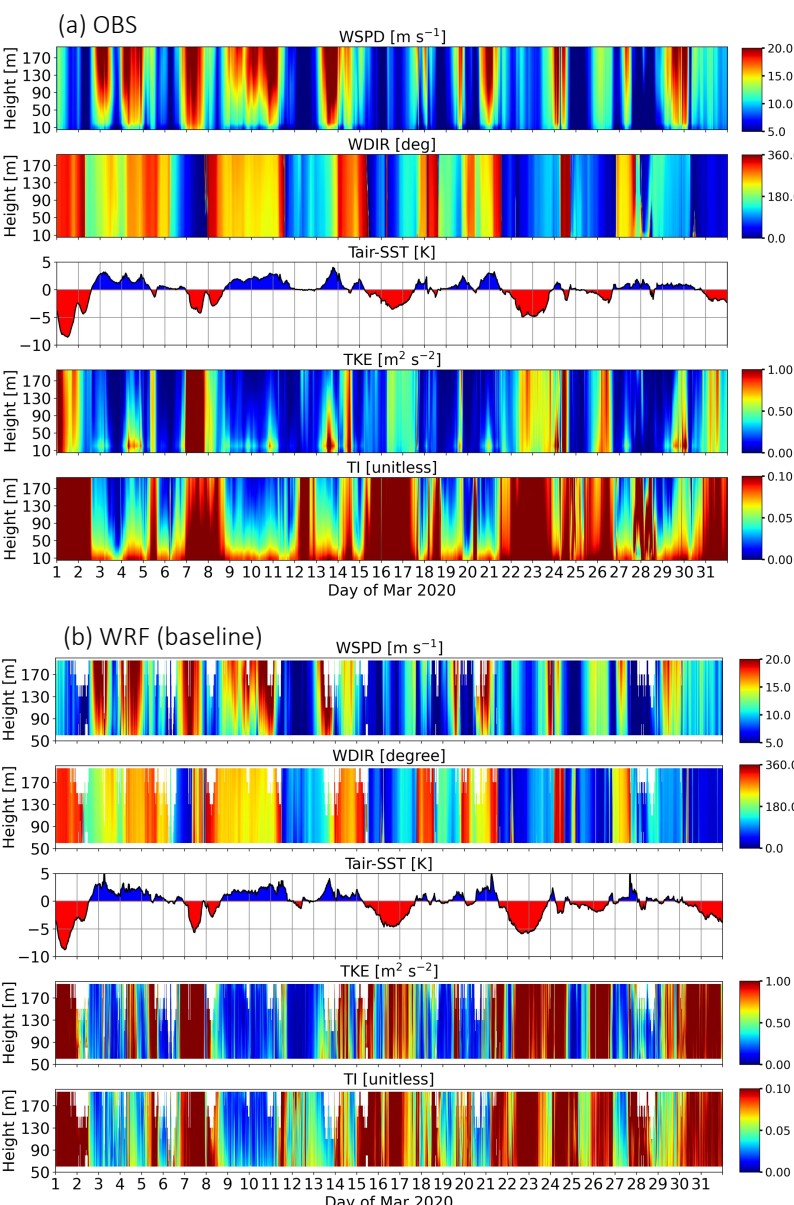

**Figure 3: Time-height comparison of wind speed (WSPD), wind direction (WDIR), air-sea temperature gradient (Tair-SST), turbulence kinetic energy (TKE), and sub-grid turbulent intensity (TI). Results for lidar and buoy observations and "baseline" simulation during the period of March are given in (a) and (b), respectively.**

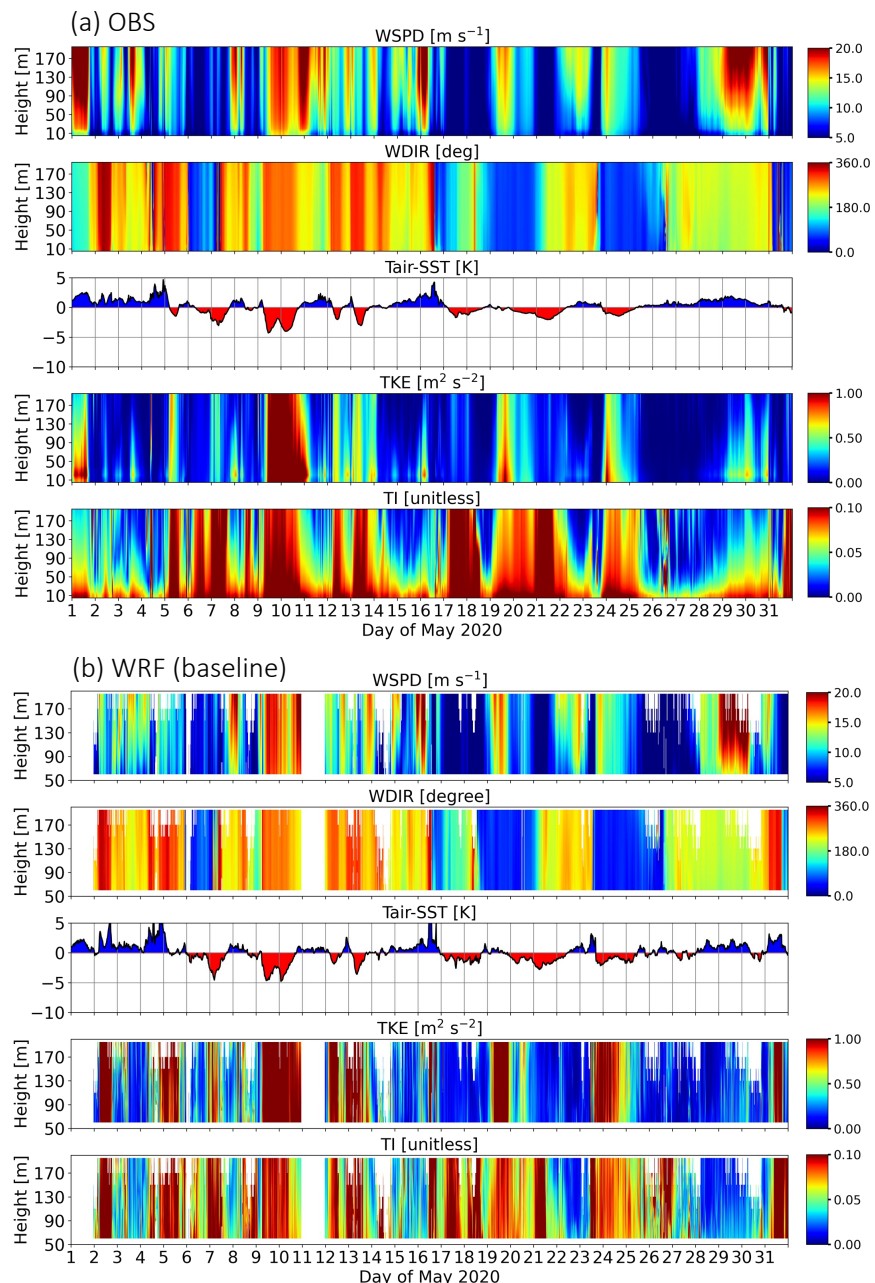

**Figure 4: Similar to Figure 3 but results for May 2020 are displayed.**

To quantify the model performance for the entire period of study, an assessment of the baseline simulation is carried out that examines the 80-m TI, TKE, WSPD, and air-sea temperature gradients. Note data points from simulations and observations are only counted when wind speed is between 5 m s$^{-1}$ and

25 m s$^{-1}$, which is consistent with turbine cut in and cut-out wind speeds of commercial wind turbines. Three metrics including root mean square error (RMSE), bias, and correlation coefficient (CC), are computed for each variable and given in Table 1. The root mean squared error (RMSE) measures how close, on average, the simulated quantities are to observations.

$$RMSE = \left[\frac{1}{n}\sum_{i=1}^{n}(y_i - o_i)^2\right]^{\frac{1}{2}} \tag{5}$$

$y$ denotes simulated result for variable of interest, $o$ represents corresponding observation. $n$ is the

number of data samples. The bias is also known as mean prediction error (MPE). It compares the simulated against observed means of the evaluated dataset:

$$bias = \frac{1}{n}\sum_{i=1}^{n}(y_i - o_i) \tag{6}$$

The correlation coefficient (CC) is used to determine the similarity between the simulated and observed data. The CC value always lies between -1 and +1, representing the range from dissimilarity to similar relation.

$$CC = \frac{\sum_{i=1}^{n}(y_i - \bar{y})(o_i - \bar{o})}{\sqrt{\sum_{i=1}^{n}(y_i - \bar{y})^2 \sum_{i=1}^{n}(o_i - \bar{o})^2}}$$

$$\bar{y} = \frac{\sum_{i=1}^{n} y_i}{n}, \bar{o} = \frac{\sum_{i=1}^{n} o_i}{n}$$

$$\tag{7}$$

Overall, the WRF model slightly underestimates TI with a bias of -0.0061 ($\sim$ 5% relative to mean value) and has a RMSE of 0.037 ($\sim$30% relative to mean value). The two main factors in computation of

TI, TKE and WSPD, exhibit notable contrasts in their assessments. While simulated TKE is underpredicted with a bias of -0.3804 $m^2 s^{-2}$ and RMSE of 0.914 $m^2 s^{-2}$, WSPD generally agrees well with lidar observation as demonstrated by relatively larger correlation coefficient (0.83) in comparison to 0.74 for TKE. This implies that the errors in TI are likely driven by the differences in TKE as the WSPD is generally well predicted. We further examine how TI and TKE errors change correspondingly with WSPD as given in Figure 5. The results confirm that, in most of the cases, large TI errors are associated with large TKE errors and have less dependency on WSPD. Lastly, the air-sea temperature gradient is reasonably represented with the highest correlation coefficient of 0.92 among all the variables.

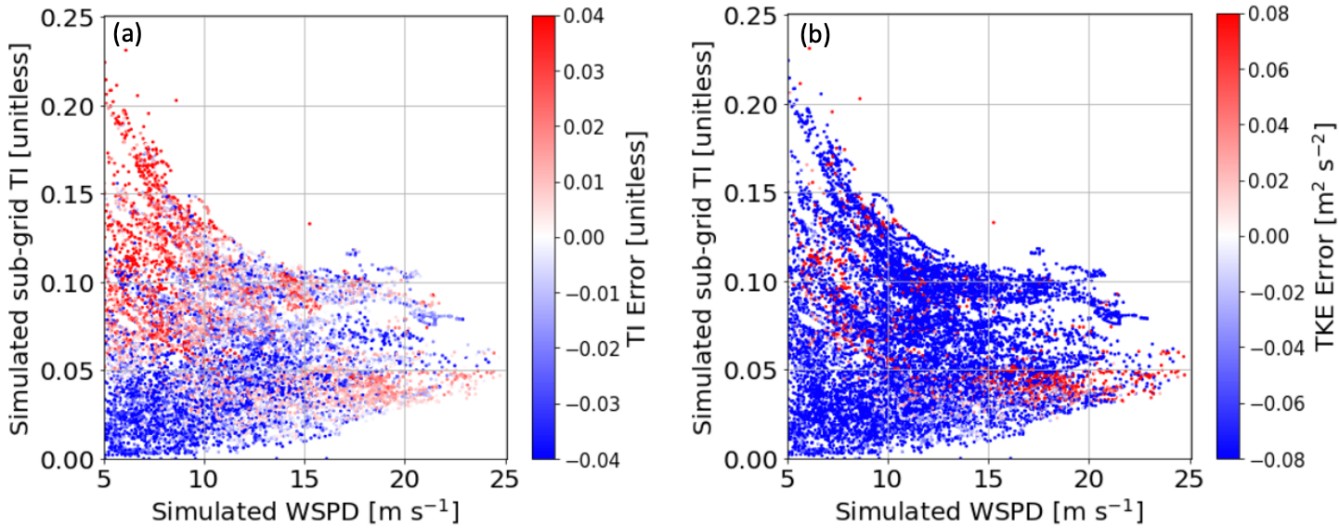

**Figure 5: Scatter plots depicting the relationship between simulated (baseline) WSPD and sub-grid TI with color shadings for representing 80-m TI and TKE errors in (a) and (b), respectively.**

**Table 1. Metrics including correlation coefficient (CC), root mean square error (RMSE), and bias, of 80-m TI, TKE, WSPD, and air-sea temperature difference are given as computed for baseline and sstupdate simulations.**

| | TI | | TKE | | WSPD | | Tair - SST | |
|---|---|---|---|---|---|---|---|---|
| | baseline | sstupdate | baseline | sstupdate | baseline | sstupdate | baseline | sstupdate |
| CC | 0.56 | 0.59 | 0.74 | 0.74 | 0.83 | 0.84 | 0.92 | 0.94 |
| RMSE | 0.037 | 0.035 | 0.914 | 0.886 | 2.37 | 2.303 | 0.921 | 0.766 |
| Bias | -0.0061 | -0.0023 | -0.3804 | -0.3526 | 0.3737 | 0.3165 | 0.3223 | 0.093 |

## 4.2 Sensitivity of SST forcing on modelled wind and turbulence

As described in Section 3.3, the default SST in the baseline simulation is replaced by the NASA JPL's SST analysis product (Chin et al. 2017) which has more fine-scale features as opposed to the SST representation in the baseline simulation. This sensitivity experiment is named as "sstupdate" and the impacts of SST forcing is examined in this section.

The same metrics used in previous section are applied to sstupdate simulation and given in Table 1. It indicates the replacement of SST forcing has positive impacts on all the examined variables. For instance, the CC, RMSE, and Bias for TI as simulated by sstupdate (baseline) are 0.59 (0.56), 0.035 (0.037), and -0.0023 (-0.0061). Similarly, the model skill improves simultaneously with respect to TKE, WSPD, and $T_{air}$-SST. This implies more realistic SST forcing helps better represent spatiotemporal evolution of stability and subsequently the surface fluxes. This then influences turbulent properties within the boundary layer such as the TKE and WSPD. A comparison of vertical profiles of TI RMSE between the two simulations (Figure 6) further shows that SST representation in the model could affect skill in TI prediction from 60 to 190 meters and error generally increases with the heights. WSPD and TKE RMSE profiles present similar trends despite the range of improvement vary with heights among the three variables. Redfern et. al. (2021) shows impact of SST replacement on wind speed modeling can be seen from 40 to 200 meters. The validation was done by comparing against lidar observations collected at three locations off the Atlantic shores from June to July in 2022.

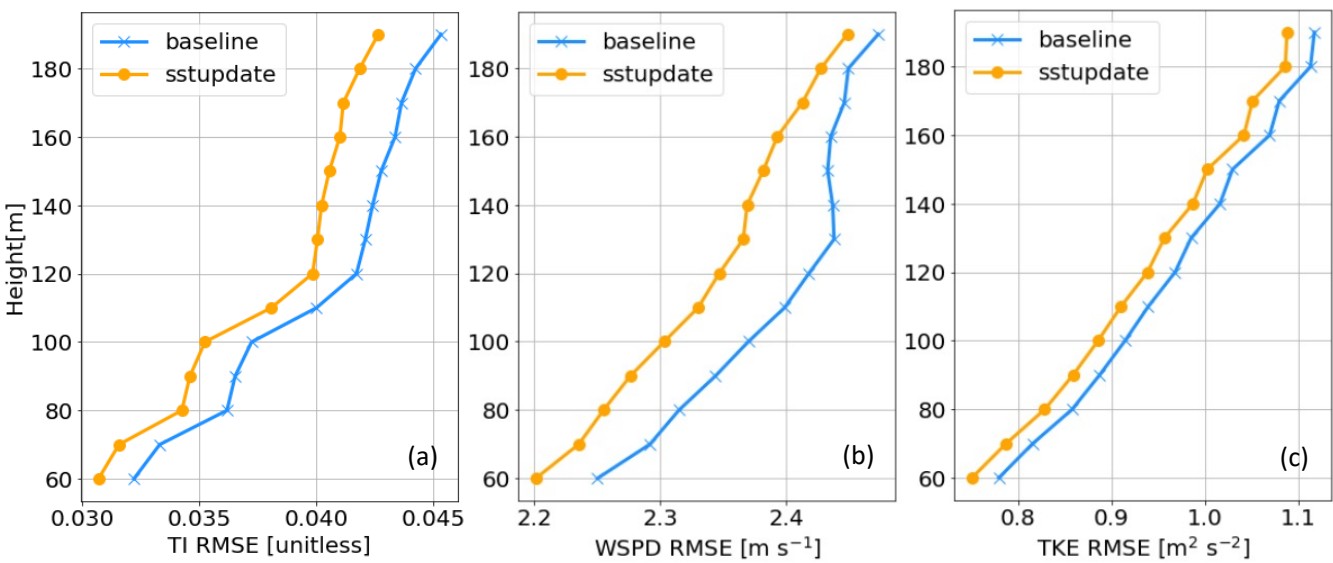

**Figure 6: RMSE profiles of (a)TI, (b)WSPD, and (c)TKE for baseline (blue) and sstupdate (orange)**
**simulations between 60 to 190 meters.**

The power density functions (PDFs) are given in Figure 7 to help describe the similarities of each variable among the three datasets. Figure 7a shows the distribution of observed TI has a wider peak between 0.025 to 0.1 than the simulated distributions. While the baseline simulation has more points with
smaller TI values (peak around 0.04), the distributions of TI from the sstupdate show an additional peak around 0.09 and 0.1 and there are more large values of TI. The median TI of sstupdate (0.066) is much closer to observations (0.07) than the baseline (0.057), indicating that the SST representation has a notable impact on TI simulation. The individual impact of SST on TKE, WSPD, and air-sea temperature difference is displayed in Figures 7b, c, and d, respectively. The model tends to produce more instances
with small TKE than was observed (0.67; Figure 7b), and sstupdate (0.4) has slightly larger median TKE than the baseline (0.377). Note part of the discrepancy in TKE may be attributed to varying uncertainty in lidar turbulence retrievals as a function of atmospheric stability. Sathe et al. (2015) found hub-height turbulence (~ 80 m) measured by pulsed Doppler lidars could be significantly higher (lower) than what is observed by a sonic anemometer during unstable (stable) atmospheric conditions. The PDFs of WSPD

in Figure 7c indicate while both simulations have relatively larger medians (10.94 and 10.86) than the observations (10.31), the additional SST information used in sstupdate slightly improve the simulations. The overall improvement in WSPD may be attributed to more accurate representation of stability as the PDF of simulated air-sea temperature gradient is improved when applying the improved SST forcing as shown by Figure 7d.

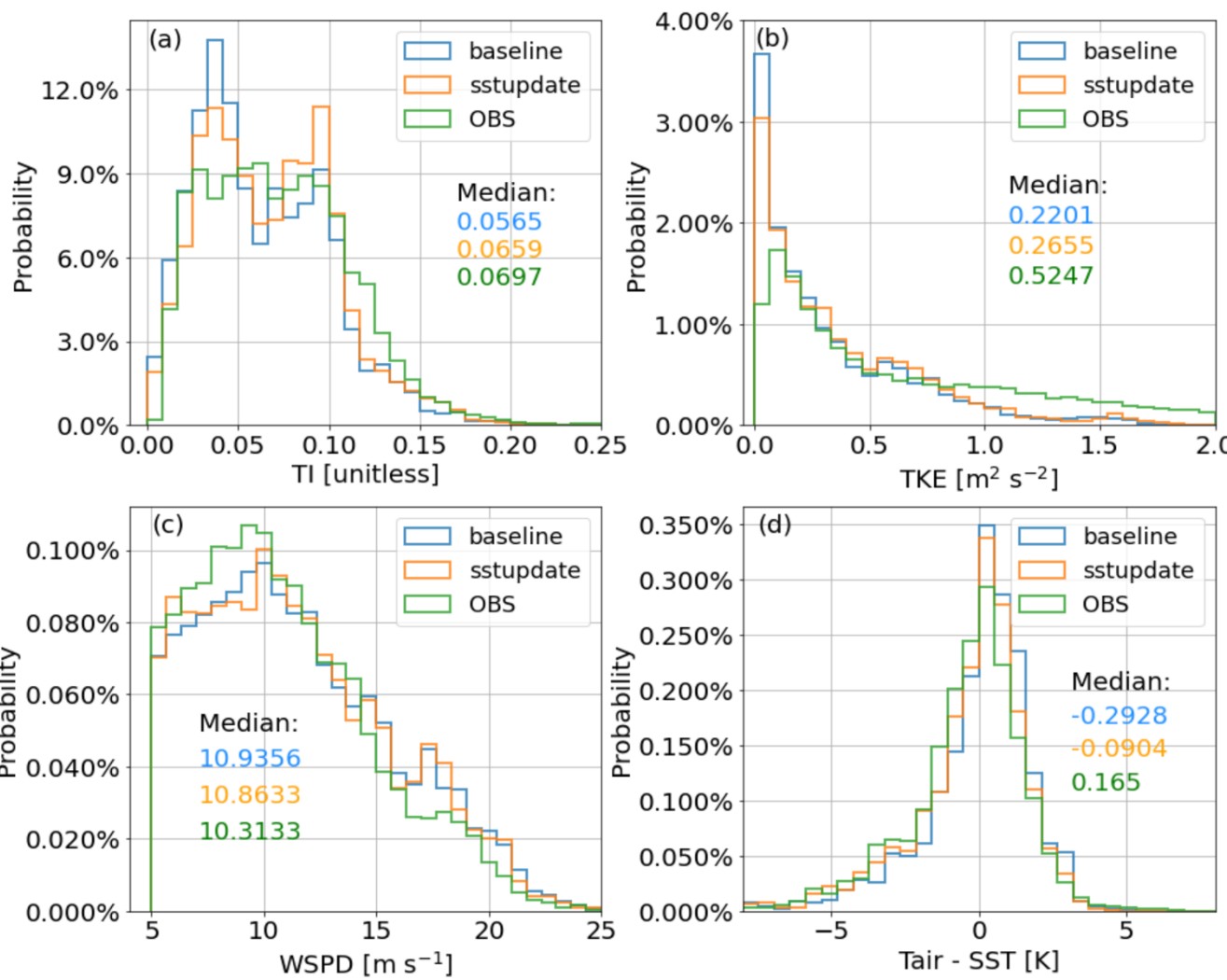

**Figure 7: Probability density function (PDF) plots illustrating the results of (a) subgrid turbulence intensity (TI), (b) turbulence kinetic energy (TKE), (c) wind speed (WSPD), and (d) air-sea temperature difference among baseline, sstupdate simulations, and observations at hub-height (~**

**80 m) though February to May of 2020. The median of each dataset is given in each panel with color**
**coded.**

Application of the higher-resolution SST impacts the time of evolution of the winds and turbulence. Figure 8 shows how the cold bias seen in the SST of baseline simulation is effectively reduced during May 2020, which subsequently fixes the cold bias in near-surface air temperature. While there are
relatively smaller differences between two simulations in the TKE and WSPD, we do find improvement in TI over some periods. For example, during May 5 – 8, 13 – 14, 17 – 18, and 30 – 31, TI simulated by sstupdate simulation are generally higher than what is simulated by baseline simulation and closer to the observed values. The correction of TKE has a larger impact on TI when WSPD is relatively small (less than 10 m s$^{-1}$), which can be explained by how TI is calculated in Eq. (4).

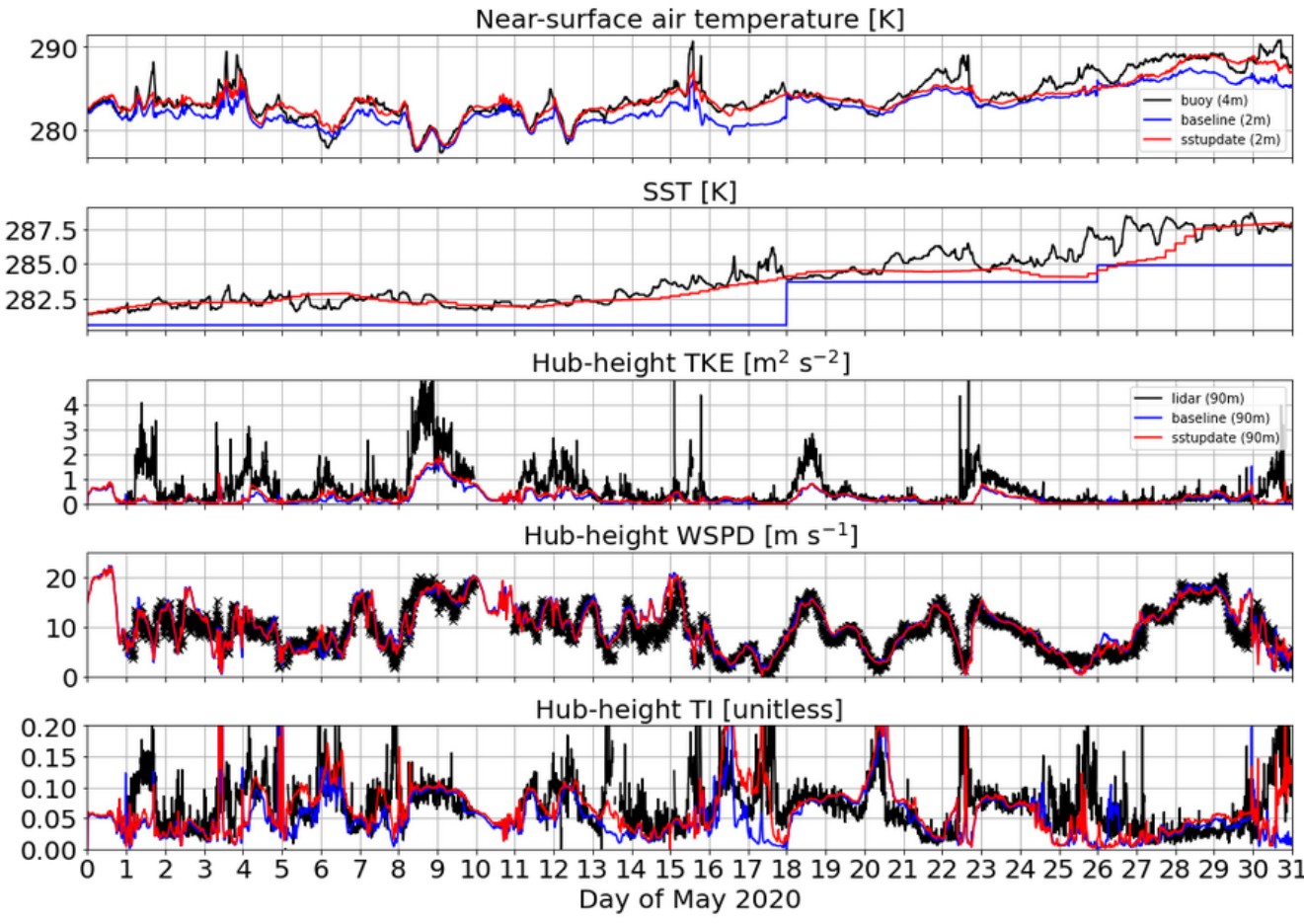


**Figure 8: Time-series display of data from baseline, sstupdate simulations and corresponding observations for May 2020. From top to bottom rows, the near-surface air temperature, sea surface temperature, hub-height TKE, WSPD, and TI are arranged.**

The impact of SST is also examined in the context of monthly variability. Figure 9 summarizes the metrics calculated for each simulation, variable, and month with the observational data as reference. The analysis suggests that the overall performance of simulated air-sea temperature gradient (TG) is improved when the higher-resolution SST forcing is used, and the improvement is more prominent in the spring months than in February. Despite only slight impact on the correlation for simulated TG by replacing the

default SST (Figure 9a), the corresponding RMSE (Figure 9b) and bias (Figure 9c) for TG are considerably reduced. While relatively small positive impact is shown for TKE and WSPD, the

improvement in the bias of TI is prominent, as shown by the reduced RMSE and bias, particularly for April and May.

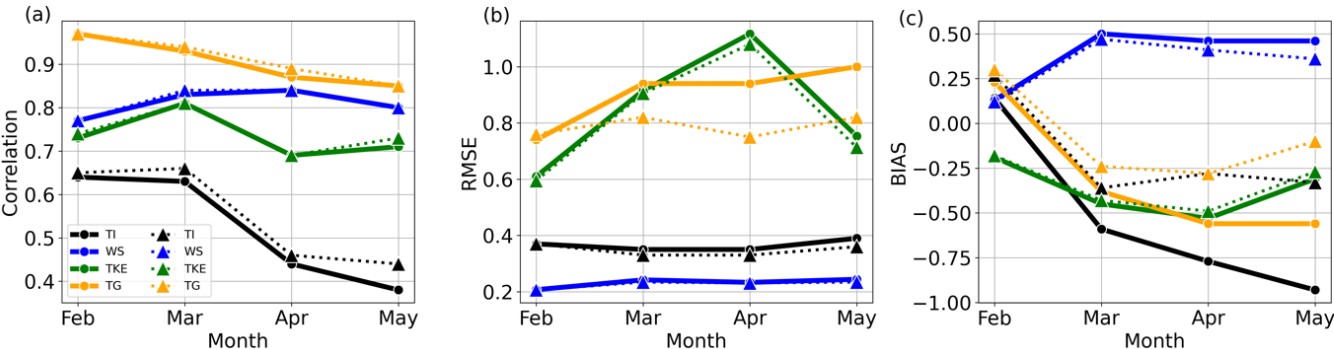

**Figure 9: Metrics of (a) correlation coefficient between the observations and simulations, (b) root mean square error (RMSE), and (c) bias for variables (TI, WS, TKE, and air-sea temperature gradient (TG)) computed against the observations for each month between February and May in 2020. The dots with solid line represent baseline result. The triangles with dashed line depict the corresponding values from sstupdate simulation.**

**4.3 Relationship between turbulence intensity and bulk Richardson number**

Many earlier studies showed that the bulk Richardson number (Rib) may be a good indicator turbulent conditions as it considers stability associated with the temperature gradient, as well as the relative contributions of buoyancy and shear (e.g. Rodrigo et al. 2015; Bardal et al. 2018; Hsu 1989; Zoumakis and Kelessis 1991; Hansen et al. 2012). The equation we use for calculating the Rib is

$$Rib = \frac{g\Delta\theta_v\Delta z}{\overline{\theta_v}(\Delta U^2 + \Delta V^2)} \tag{4}$$

where g denotes gravitational acceleration, $\Delta\theta_v$ is virtual potential temperature difference across a vertical layer of thickness $\Delta z$, $\Delta U$ and $\Delta V$ represent the vertical gradient in horizontal wind components. The virtual potential temperature gradient ($\Delta\theta_v$) is computed by using the air temperature (4 m) and sea surface temperature measurements on the buoy. The vertical gradients of horizontal wind components ($\Delta U$ and $\Delta V$) are obtained by using wind measurement at 100 m from the Doppler Lidar (DL) and 4 m from the

buoy. We also applied a similar approach using the WRF model output. Note the wind and temperature

gradients are not computed from the same heights, and the bulk Richardson number calculated here will

only be used to inform stability qualitatively (Howland et al. 2020). As before, data for cases where the

wind speed was less than 5 m s⁻¹ and greater than 25 m s⁻¹ have been removed as described earlier. Figure

10a displays the calculated Rib in timeseries during February 2020 as an example. It shows that both

experiments reproduce the occurrence of observed events with unstable conditions (large negative Rib

values). For instance, in February 14 – 15, 19 – 20, 21, the model was able to produce large TI (over 0.2,

Figure 10d) when Rib approaches -10 (Figure 10a).

While simulated results show that the model has good skill for large TI (over 0.2) events, there are

periods with moderate TI (0.05 to 0.2) where the model misses the observed peaks as indicated by the

green arrows in Figure 10d. It is found in those time periods, the buoyancy component ($\Delta\theta_v$) are mostly

near zero (neutral conditions) or even positive (stable), whereas the shear component ($\Delta U^2 + \Delta V^2$) could

be more variable from time to time, indicating the model may have less skill in TI prediction when the

buoyant forcing is weak.

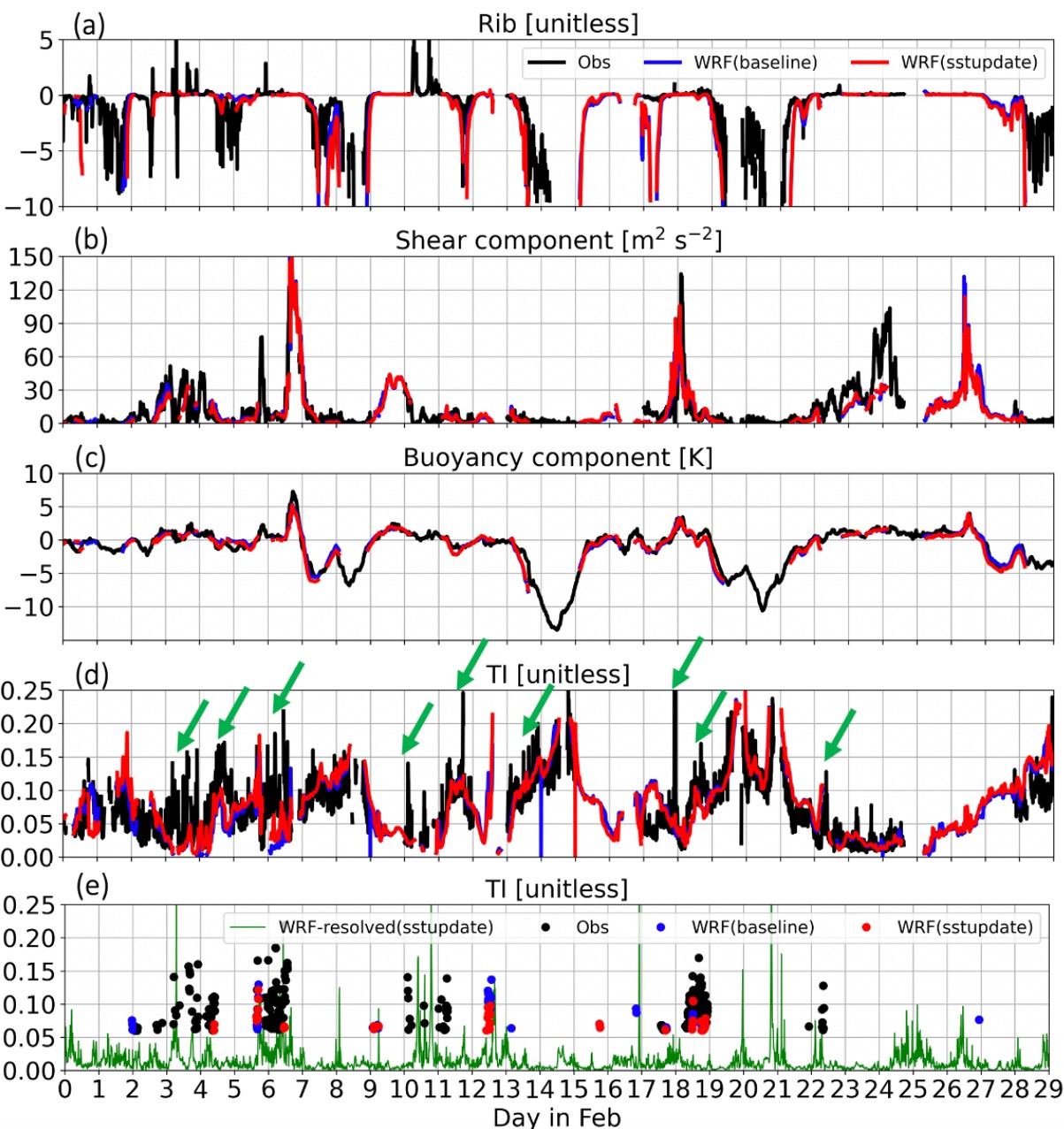

**Figure 10:** Timeseries plots of observed and simulated (a) bulk Richardson number, (b) shear component (m² s⁻²), (c) buoyancy component (K), and (d) sub-grid turbulence intensity (TI) during February 2020. Green arrows in (d) indicate events with large contrast in TI between model and observation. Filtered data points of sub-grid TI by using the thresholds of buoyancy > 0 K, shear < 20.0 m² s⁻², and TI > 0.06 are denoted in (e). Green line represents the model-resolved TI.

To explore how TI can be related to the buoyancy and shear components of Rib, TI is mapped in regard to the two terms (Figure 11). Figure 11a shows the observed TI is generally larger during periods when conditions are unstable and there is a large magnitude of negative buoyancy component (SST is larger than air temperature). It implies that in convective regimes (buoyancy component < 0), more vigorous turbulence leads to stronger vertical mixing which significantly reduces the wind shear. On the

other hand, in stable regimes (buoyancy component > 0), less turbulent mixing often results in larger values of wind shear.

        Although in general, TI decreases as the buoyancy becomes more positive and the atmosphere becomes stable as identified in the observations and the two WRF model configurations (Figures 11a, b, and c), a population of reddish circles (large values of TI) in the lower-right quadrant of the figure does

not follow this relationship (Figure 11a). While both simulations fail to represent these cases (Figures 11b and c), the fractional difference of simulated TI between the two simulations indicates that the sstupdate simulation generally has larger TI than the baseline simulation (Figure 11d). This can be attributed to overall reduction of cold bias in the baseline simulation as shown in Figure 9. Furthermore, a larger fractional increase of TI is found in the regime where the conditions are between neutral and slightly

stable (buoyancy component >= 0). This is most likely due to the weak negative temperature gradient in baseline simulation becoming positive after replacing the SST forcing. Despite the correction, large TI is rarely simulated in the lower-right regime. This result is not surprising as the formulation in the MYNN parameterization does not allow large values of TKE to be diagnosed in stable conditions.

        The next step was to locate these specific data points in time by applying thresholds of buoyancy

component > 0 K, shear component < 20.0 $m^2$ $s^{-2}$, and TI > 0.06 (Figure 11a). It shows that features leading to the large values of TI in the observation and simulations may not overlap in time as the model may fail to simulate realistic conditions (Figure 10e). Nevertheless, it is found that many of these cases with positive buoyancy component and large TI are aligned with periods that have notable differences between observed and simulated TI, as denoted by green arrows in Figure 10d. Therefore, we conclude

that the underrepresentation of this regime is likely responsible for the majority of the model bias in TI.

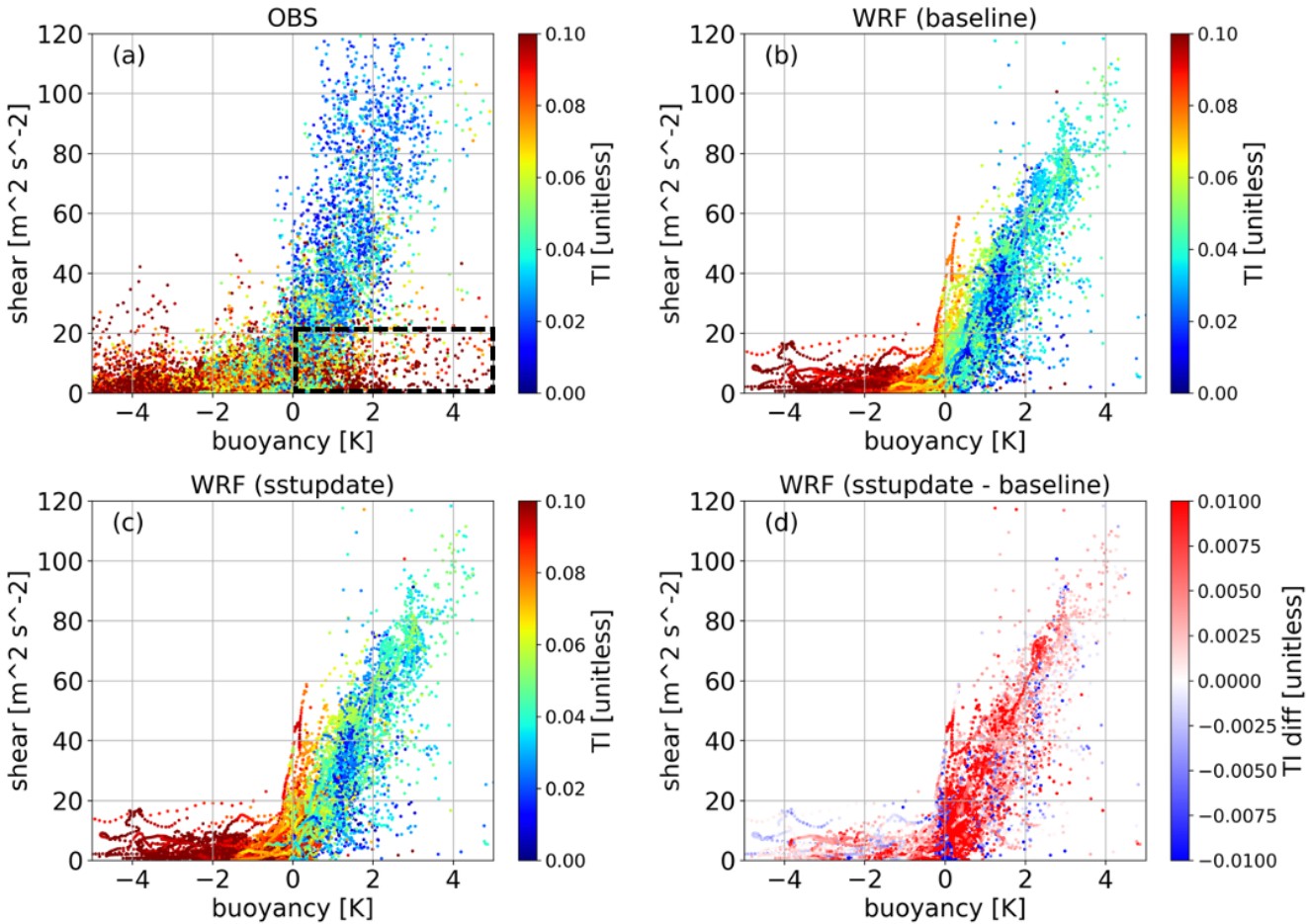

**Figure 11: Scatter plots mapping the hub-height sub-grid TI in the coordinates of buoyancy ($\Delta\theta_v$) and shear ($\Delta U^2 + \Delta V^2$) components for the entire period of study. Results of (a) observation (OBS), (b) baseline simulation, (c) sstupdate simulation, and (d) percentage of fractional difference (%) between sstupdate and baseline simulations are displayed. The black dashed-line box in (a) indicates the regime that is used to extract the data points as shown in Figure 10e.**

While some of the intermittent turbulent events may cause the occurrence of large TI under small shear and stable conditions, we find that most of the cases with large amounts of TI in stable conditions and weak shear can be linked to fluctuations in wind speed associated with mesoscale features of the flow and they are partially resolved by mesoscale model, or for which there are timing errors. In Figure 10e,

the mesoscale (model-resolved) TI, described in Section 3.2, is denoted by the green line. For the periods such as Feb. 3 – 4, 6 – 7, 10 – 11, when the simulated sub-grid TI is much smaller than observation, spikes in mesoscale (model-resolved) TI have amplitudes similar to the observed values. This suggests the need to consider both sub-grid and mesoscale TI when modelling TI derived from a mesoscale atmospheric

model. Furthermore, the uncertainties in the timing of simulated mesoscale weather events may lead to notable contrasts between the timing of modelled and observed peaks in TI. This can impact the simulated TI in two ways as the mesoscale activity not only generates variance in the winds over larger spatiotemporal scales than the sub-grid turbulence, but also effectively offsets dynamic and thermodynamic conditions near the sea surface in the boundary layer that influence the simulated sub-

grid TI generated by the boundary-layer parameterization. Our analysis demonstrates the utility of identifying scale-dependent uncertainties of TI modelling, which allows us to isolate the causes of errors in the simulated TI.

The metrics used in Table 1 are employed again to assess the impact of resolved TI on the TI prediction and listed in Table 2. Results indicate the addition of resolved TI reduces the bias for baseline

configuration (-0.0061 to 0.0017) but enlarges the bias in the sstupdate configuration (-0.0023 to 0.0055). Moreover, there is worse correlation and RMSE when both sub-grid and resolved TI are considered. For instance, the correlation coefficients are reduced from 0.56 to 0.53 and 0.59 to 0.56 for baseline and sstupdate configurations, respectively. Considering the effects of the higher-resolution SST forcing and resolved TI can lead to an overestimate of the TI, as the bias changes sign and the RMSE increase slightly

compared to the sub-grid TI only. We suspect this could be associated with the uncertainties in simulating mesoscale events as depicted in Figure 10e. The events with larger resolved TI are not necessarily coincident with observations, which explains why simply summing up sub-grid and resolved TI in time may introduce additional errors in the simulated TI.


**Table 2. Metrics including correlation coefficient (CC), root mean square error (RMSE), and bias, for TI that are computed based on total (sub-grid and resolved) TI simulated by baseline and sstupdate simulations.**

|  | baseline | sstupdate |
|---|---|---|
| **CC** | 0.53 | 0.56 |
| **RMSE** | 0.039 | 0.038 |
| **Bias** | 0.0017 | 0.0055 |

## 5 Summary and conclusion

Algorithms to derive TI are successfully implemented in the WRF model and tested offshore using multi-month data set collected at MVCO using a combination of Doppler lidar, tower, and buoy data. Simulated TI is divided into two components depending on scale, including sub-grid and grid resolved. To obtain simulated TI, we calculate the square root of sum of horizontal wind variances and then divide it by the mean wind speed over a 10-min window. While sub-grid TI is diagnosed from parameterized turbulence kinetic energy (TKE) through the MYNN PBL parameterization, resolved TI is estimated by using the model-resolved wind variances.

The modelled TI computed over a wide range atmospheric conditions are analyzed and validated by using a variety of observations collected at the offshore tower at MVCO between February and May in 2020. The primary findings include:

- The model's PBL scheme reacts reasonably well to changes in the vertical temperature gradient near sea surface despite the fact that the simulated TKE is generally smaller than what is observed by the lidar, especially for events with large TKE values.
- The modified WRF model slightly underestimates TI, and the error is mainly attributed to relatively large negative bias in TKE as the predicted wind speed generally agrees with observation.

- The WRF model has difficulty predicting periods of weak TI due to smaller air-sea temperature differences and fluxes within the simulated boundary layer.

- Overall cold bias in the SST of baseline simulation is effectively reduced by substituting it with more accurate SST forcing. It subsequently reduces model biases in near-surface air temperature as well as hub-height WSPD and TI.

- A regime of large observed values TI during periods with positive buoyancy and weak shear is identified, but are not captured by the model. Many of the events occurred in conjunction with mesoscale weather systems, but directly summing up sub-grid and resolved TI does not improve the TI prediction. This is because the primary source of uncertainty in those events are caused by the unrealistic representation of mesoscale weather systems, including timing errors, in the model.

Our analysis suggests additional model constraints are required to further improve model representation of TI and TKE in mesoscale or finer-scale cloud (-system) processes. Approaches that couple mesoscale model with data assimilation techniques to improve skill in the simulated mesoscale flow features and cloud predictions (Tai et al. 2020, 2021; Gaudet et al. 2022) could be very beneficial.

**Code availability**

The codes implemented in the WRF model for the presented simulations as well as the scripts for data analysis can be made available up request. In case of interest, please contact the authors.

**Data availability**

The observational and simulated datasets used in this study can be made available upon request.

**Author contribution**

ST: conceptualization, methodology, execution, analysis, and writing. LB: conceptualization, methodology, supervision, and review. RK: observational data processing, consulting, and review. RN: observational data processing. AK: observational data consulting and review.

## Competing interest

The contact author has declared that neither of the authors has any competing interests.

## Acknowledgment

Pacific Northwest National Laboratory is operated by Battelle for the U.S. Department of Energy under Contract DE-AC05-76RLO1830.

## Financial support

This research was supported by the National Offshore Wind Research and Development Consortium under agreement number 147502.

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
