# Peer review of "Validation of Turbulence Intensity as Simulated by the Weather Research and Forecasting Model off the U.S. Northeast Coast"

_Wind Energy Science, 2022_

## Author Comment (AC1)

The manuscript "Validation of Turbulence Intensity as Simulated by the Weather Research and Forecasting Model off the U.S. Northeast Coast" explores the turbulence intensity reproduced with WRF model offshore, comparing with observations from a lidar profiler and other instrumentation. The manuscript is worth to be published, contributing to a better knowledge of the ability of the model to resolve the main meteorological variables and the TKE, or the turbulence intensity. Results are well presented, plots are correct and illustrate well the results. However, in the text, in general, my feeling is that more detailed explanations about the obtained results are needed. In addition, 3 major points need to be addressed.

My major concerns are:

- Results with SST update are very interesting and seem to be closer to observations. It seems that a big part of the paper is about the discussion about the added value of the SST update in comparison with the baseline simulation. Have you considered focusing the manuscript in these topic, and maybe changing the title, the main focus, etc. In the present form it is confusing why the sst update is a big topic and not others, such as the PBL scheme, the surface layer scheme, initial conditions, vertical levels... etc

**Response:** We thank the reviewer for the great comment. We would like to note that the choices of model physic parameterizations in this study are consistent with the setup for the 20-year wind resource dataset released by the National Renewable Energy Laboratory (NREL) (Optis et al. 2020) The model configuration for the CA20 dataset was determined through a series of sensitivity studies including varying PBL schemes, reanalysis forcing, sea surface temperature forcing, and surface layer schemes.

Since the HRRR analysis, the atmospheric forcing we used for our WRF simulations, does not include SST data, we must adopt one from the existing SST products. The NASA JPL GHRSST data was chosen as it has higher resolution (0.01°) than what were used in Optis et al. (2020). A few sentences are added in Section 3.3 from Line 291 to 302 to provide more details on this.

While the discussion of sensitivity of prescribed model SST is particularly included in the manuscript, we acknowledge many other treatments in the model would interactively contribute to the uncertainties in simulated wind and turbulence at the hub height. Hence, we briefly describe the potential sources of model uncertainty in Section 3.1 from Line 139 to 141 to provide more background and hopefully it helps clarify why we didn't test the sensitivity to model physics or other factors.

- Metrics are included but not described in methodology. Please, include a section with the formula of Bias, RMSE and others if needed.

**Response:** We thank the reviewer for the comment. To address this, we've included descriptions for the metrics (RMSE, bias, and correlation coefficient,) in the Section 4.1 from Line 357 to 365.

-Another important point is what was the criteria to choose the physics WRF configuration. Are there previous studies that proof a suitable model physics? Apart from SST update, have you tried any other experiment varying physics? PBL? Other initial conditions? Vertical levels?

**Response:** We thank the reviewer for bringing up these questions. As in our response for the first comment, the model configuration was based on the results of model sensitivity experiments described by Optis et al. (2020). We acknowledge many other treatments in the model could contribute to the uncertainties in simulated wind and turbulence at the hub height. We've added a few sentences from Line 144 to 157 in Section 3.1 to address the comment:

"The choices of model physics parameterizations for this study are consistent with the setup for the 20-year wind resource dataset released by the National Renewable Energy Laboratory (NREL) (Optis et al. 2020). Optis et al. (2020) conducted a series of model sensitivity experiments with respect to surface layer and PBL parameterizations, reanalysis data, and SST forcing. The results of their model assessment indicated that the largest uncertainty is associated with the choice PBL parameterizations, and the Mellor-Yamada-Nakanishi Niino (MYNN) boundary layer parameterization (Nakanishi and Niino 2009) generally outperforms the Yonsei University (YSU, Hong et al. 2006) scheme off the east coast of North America. Hence the MYNN boundary layer parameterization, as well as other parameterizations described by Optis et al. (2020) were used for generating the CA20 dataset as well as our simulations."

Some minor comments are indicated below:

Line 45-46: and during night?
**Response:** The structure of marine PBL is very dependent on both the variations of sea surface temperature and air temperature near the sea surface. We revised the sentence to clarify this point as in Line 43 and 44.

Line 109: 23 dB (include an space)
**Response:** This is corrected.

Line 146: w_variance is always negligible? Please, detail cases when this assumption is not correct.
**Response:** We reviewed the observational datasets collected at the MVCO ASIT and we used them to characterize the distribution of observed wind variances and summarized results as following:

Figure below shows the fraction of three components of wind variances normalized by the total variance ($\frac{\sigma_u^2}{\sigma_u^2+\sigma_v^2+\sigma_w^2}, \frac{\sigma_v^2}{\sigma_u^2+\sigma_v^2+\sigma_w^2}, \frac{\sigma_w^2}{\sigma_u^2+\sigma_v^2+\sigma_w^2}$) as a function of stability (using buoy-measured air temperature minus SST as its proxy). The results suggest overall, the fractions of horizontal wind variances ($\sigma_u^2$ and $\sigma_v^2$) are larger than the vertical wind variance ($\sigma_w^2$). In addition, there is no evident correlation between the fraction of $\sigma_w^2$ and stability. In most of the conditions, the fraction of $\sigma_w^2$ is no greater than 0.2. Only a few data points exceed 0.2 but they mostly occur during neutral conditions (Tair – SST is close to zero).

[Figure]

We further analyze what may cause those data having relatively large fraction of $\sigma_w^2$. By looking at the corresponding horizontal wind variances ($\sigma_u^2$ and $\sigma_v^2$) in function of fraction of $\sigma_w^2$ as in the figure below, we find that the relatively large fraction of $\sigma_w^2$ (larger than 0.4) is most likely due to concurrently small values in $\sigma_u^2$ and $\sigma_v^2$ (smaller than 1 m$^2$ s$^{-2}$).

[Figure]

To address this comment, we add a figure (Figure 2 in the revised manuscript) to illustrate the observed PDFs of three wind components of variances and their dependency on stability. Corresponding descriptions can be found in Section 3.2 from Line 202 to 206.

Table 1: Mean wind speed in 10-min window
**Response:** This is corrected.

Fig2 caption: observations from lidar? Please indicate
**Response:** The observations are from both lidar and buoy. It is now included in the caption.

Line 218: is there cancellation in the MB between negative and positive values?
**Response:** From the formula of bias, the negative and positive values could possibly compensate each other. However, if the sign of biases is relatively consistent throughout the data sample, MB would be still useful in identifying the overall bias.

Line 286: relative importance? Please, give details
**Response:** We revised the sentence now as "which can be explained by how TI is calculated in Eq. (4)."

Figure 7: green line in panel WSPD is difficult to see
**Response:** The green line in this figure (Figure 8 in revised manuscript) is changed to blue.

Line 322: please explain better
**Response:** The sentences mentioned here are revised.

Line 339: please define the buoyancy component
**Response:** We modified the sentences in Line 514 and 515 where the two terms buoyancy and shear components are first defined in the text.

Line 354: yes, so what about YSU?
**Response:** In our responses to the major comment, we clarify the appropriateness of using the MYNN parameterization for our study. In addition, since TKE is not diagnosed in the module of the YSU scheme, it is difficult to answer the question at this moment.

Line 361: this regime can be low shear and stable conditions, maybe intermittent turbulent events?
**Response:** It is possible that intermittent turbulent events may cause occurrence of large TI during low shear and stable conditions in addition to resolved turbulence linked to mesoscale convective events. To address this, we revised the sentence in Line 580.

Line 370: ok, but also intermittent turbulent events?
**Response:** Please see the response for previous comment.

Figure 10 caption: buoyancy is the numerator of Rib?
**Response:** The buoyancy here is defined as air-sea virtual temperature difference $(\Delta\theta_v)$ as stated in Line 514. We include exact ingredients for both buoyancy and shear components in the caption to clarify.

Table 2: please, join it with table 1 or refer to it in the discussion
**Response:** The Table 2 is already referred in Line 604.

---

## Author Comment (AC2)

Review WES-2022-84: Validation of Turbulence Intensity as Simulated by the Weather Research and Forecasting model off the U.S. Northeast Coast, by Tai et al.

This paper makes use of WRF model outputs and calculates turbulence intensity (TI) in terms of a standard WRF output Turbulence Kinetic Energy (TKE) (Eq. 3). This study also made efforts to improve the simulation by using alternative Sea Surface Temperature (SST). The comparison of TI with measurements from one site suggests there is agreement between measured and modeled values, particularly for TKE, wind speed and temperature parameters. However, even though there is slight improvement in TI using new SST in the modeling, the results of TI are not as good as the other parameters which are direct outputs from the model.

The reviewer believes that it is not a coincidence, as the reviewer is not convinced the applied algorithm for calculating TI using TKE is correct. This is seen as a major point, even though there are several merits in the current study, including the clear paper structure, sensitivity tests of SST and corresponding analysis. The reviewer therefore recommends 'Major revision' (or even 'rejection', depending on the editor's judge how serious the following point 1 is).

- The authors simplified Eq. (1) and Eq. (2) to Eq. (3) by stating: "Here we assume sigma_w^2 is negligible as the sigma_w^2 is often much smaller than sigma_u^2 and sigma_v^2 offshore due to relatively low surface flux".

Firstly, surface flux of what? Is it always small? Probably not when you later address "unstable" conditions.

**Response:** We thank the reviewer for his/her insights regarding uncertainties in TI modeling. Here, we meant to say surface "sensible heat" flux. The sea surface sensible heat is extracted from the ocean in association with an air-sea temperature difference. In this case, it is apparent the value of latent heat flux would vary with the magnitude of air-sea temperature difference and is used in this study as the proxy of stability. We have revised the paragraph to avoid confusion. Please refer to discussions in Section 3.2.

To verify dependency of wind variances on corresponding stability, we have conducted observational analysis and please find more detailed discussions in the third response below. We note the algorithm for sub-grid TI calculation used in our study does include the vertical wind variance ($\sigma_w^2$) as the formula used in this study (Eq. 4 in the manuscript) employs TKE in the numerator:

$$TI \cong \frac{\sqrt{2 * TKE}}{\overline{U}}$$

In this case, the three-dimensional wind components of variances are considered in our algorithm. To clarify this, we revised the relevant paragraph in Section 3.2 from Line 196 to 261 and hope it addresses the comment.

Moreover, as described in Section 4.3, the model-resolved (mesoscale) TI is also considered in our study. Our analysis shows it may also contribute to a large fraction of uncertainty in modeled TI when mesoscale systems occur.

Finally, potential uncertainty in lidar-measured turbulence is also raised by one of the reviewers. The reviewer suggests including the conclusion proposed by the report: Sathe, A., Banta, R., Pauscher, L., Vogstad, K., Schlipf, D., Wylie, S., 2015. Estimating Turbulence Statistics and Parameters from Ground- and Nacelle-Based Lidar Measurements. IEA Wind Task 32 Expert Report. ISBN 978-87-93278-35-6. Their results indicated that pulsed lidars can measure a value of turbulence which is significantly larger than a sonic anemometer at 80 m above the ground under unstable conditions. We have included relevant discussion in Section 4.2 from Line 421 to 424.

Secondly, here the authors also "followed" "Eq. 1 in Bodini et al 2020" - which only has the first part of Eq. (3) and which was used for lidar measured 2-min averaged wind, which is very different from the WRF output here.

**Response:** We discussed the limitation in obtaining three-dimensional winds on turbulent scales from the model parameterization scheme in Section 3.2. This prevents us from using original form of TI formula (Eq. (1) in the manuscript). And since the simulated hub-height TI are validated by using lidar measurements, the current derivation should be reasonable. To address the reviewer's comment, we have revised the paragraph from Line 196 to 261 and included two additional references (Shaw et al. (1974) and Wharton and Lunquist (2011)) that used the same equation as Eq. 3 in the manuscript for TI calculation from lidar measurements.

Thirdly, one has to prove if $sigma\_w^2$ is negligible – one cannot simply assume. As $sigma\_u$, $sigma\_v$ and $sigma\_w$ are boundary turbulence parameters and there are numerous literatures addressing the relationship between the three variables in the surface layer when being normalized with friction velocity. A recent report from DTU (Larsén, X. G. (2022). Calculating Turbulence Intensity from mesoscale modeled Turbulence Kinetic Energy. DTU Wind Energy. DTU Wind Energy E No. E-0233) derived the relation between TI and TKE using boundary-layer turbulence model, the Kaimal model. From this approach, $sigma\_w$ is not negligible. It is recommended to refer to this report.

**Response:** We thank the reviewer for suggesting an alternate approach in deriving TI from TKE. The derivation in Larsen (2022) is based on a coordinate system with the perspective of a wind turbine, meaning that the u-component considered in its TI equation (Eq. 1) is streamwise (along) wind component, while we consider the full three-dimensional wind here. The differences in the definitions makes it difficult to compare the two approaches. In addition, it is known that the Kaimal model used in the Larsen (2022) study is more suitable for application in neutral conditions. This may potentially limit the applicable scenarios for our modeling. Therefore, we have included the suggested reference with a brief description in Section 3.2 from Line 196 to 201, but didn't apply the approach to our study.

To address the comment in relative importance of wind variances, we conducted additional analysis of observational datasets collected at the MVCO ASIT and we used them to characterize the distribution of observed wind variances and summarized results as following:

Figure below shows the fraction of three components of wind variances ($\frac{\sigma_u^2}{\sigma_u^2+\sigma_v^2+\sigma_w^2}$, $\frac{\sigma_v^2}{\sigma_u^2+\sigma_v^2+\sigma_w^2}$, $\frac{\sigma_w^2}{\sigma_u^2+\sigma_v^2+\sigma_w^2}$) normalized by the total variance as a function of stability (using buoy-measured air temperature minus SST as its proxy). The results suggest overall, the fractions of horizontal wind variances ($\sigma_u^2$ and $\sigma_v^2$) are larger than the vertical wind variance ($\sigma_w^2$). In addition, there is no evident correlation between the fraction of $\sigma_w^2$ and stability. In most of the conditions, the fraction of $\sigma_w^2$ is no greater than 0.2. Only a few data points exceed 0.2 but they mostly occur during neutral conditions (Tair – SST is close to zero).

[Figure]

We further analyze what may cause those data having relatively large fraction of $\sigma_w^2$. By looking at the corresponding horizontal wind variances ($\sigma_u^2$ and $\sigma_v^2$) in function of fraction of $\sigma_w^2$ as in the figure below, we find that the relatively large fraction of $\sigma_w^2$ (larger than 0.4) is most likely due to concurrently small values in $\sigma_u^2$ and $\sigma_v^2$ (smaller than 1 m$^2$ s$^{-2}$).

[Figure]

A figure (Figure 2 in the revised manuscript) is added to illustrate the observed PDFs of three wind components of variances and their dependency on stability. Corresponding descriptions can be found in Section 3.2 from Line 202 to 206.

- The authors suggest "for any given value of total horizontal wind variances, TI would be larger when the wind speed is relatively smaller or vice versa".

This is not true. The authors suggest TKE, and U are not correlated, which is not correct, particularly over water. Stronger winds over water generally lead to rougher surface (if the water surface is not covered by foam), which corresponds to larger TKE.

**Response:** We thank the reviewer for pointing out this. We want to note that the description mentioned in the comment was not meant to imply TKE and wind speed are not correlated but just an explanation of the equation itself.

To verify the correlation between TKE and wind speed, we plotted the hub-height wind speed and TKE as collected by the MVCO ASIT lidar from January to mid-June of 2020. While it suggests larger TKE values can be observed as wind speed increases, the spread of TKE values is large and there is a cluster of points with relatively small hub-height TKE even when the wind speed is relatively large.

[Figure]

Moreover, in the manuscript, we also discussed how the TI model errors can be correlated with wind speed and TKE as given in Figure 5. The results indicate large TI errors are associated with large TKE errors and have less dependency on wind speed.

Despite the relationship between TKE and TI is not fully linear, we do agree that the original interpretation may be somewhat misleading. Hence, we eliminated and revised the corresponding paragraph. Hope this helps answer the reviewer's comment.

- Would the authors please explain how much value is added when downscaling HRRR (3 km) to WRF (2 km)?

**Response:** We thank the reviewer for the question, and we are happy to provide additional clarification. We need to perform model downscaling (or initialization) by using any one of the valid atmospheric analysis products. Compared to other available reanalysis products (e.g., ERA5, MERRA2, and FNL), HRRR analysis has several advantages including 1) the model core of HRRR, the WRF model, is identical with what we use in this study; 2) it has a grid spacing of 3 km, which is very close to what we use (2 km); 3) it is constrained hourly by assimilating radar observations including Doppler velocity and reflectivity. The constrain in simulated hydrometeors is unique among all the analysis products which reduces some of the uncertainties in the prediction of precipitating clouds. Therefore, we decided to use HRRR analysis as the initial and boundary conditions for our simulations. A few sentences are added in Section 3.1 from Line 163 to 169 to supplement.

- In some of the analysis, the authors mixed "shear" with "TI".

**Response:** We went through the manuscript and fixed it as much as we can find.

**Reference:**
Sathe, A., Banta, R., Pauscher, L., Vogstad, K., Schlipf, D., Wylie, S., 2015. Estimating Turbulence Statistics and Parameters from Ground- and Nacelle-Based Lidar Measurements. IEA Wind Task 32 Expert Report. ISBN 978-87-93278-35-6.

---

## Author Comment (AC3)

The study represents an interesting evaluation of the ability of the WRF model to predict offshore turbulence intensity. It makes a valuable contribution, particularly with regard to the validity of combining sub-grid and resolved TKE quantities to estimate TI. There are some areas that should however be addressed before publication:

1) The paper should make it clear that it is focusing on the sensitivity of the results to the use of higher resolution SST data as presumably different PBL schemes will have quite an effect on accuracy, particularly under different stability conditions.

**Response:** We thank the reviewer for the constructive comment. The choices of physical parameterizations for our simulations are essentially inherited from the setup for the 20-year wind resource dataset released by the National Renewable Energy Laboratory (NREL) (Optis et al. 2020) as the model configuration was determined through a series of model sensitivity experiments. That's the reason why the online TI calculation was implemented within the module of MYNN PBL parameterization in WRF model.

While the sensitivity of prescribed SST data in the model is specifically addressed in the manuscript, we acknowledge there are many other factors including the choice of PBL scheme would give variable results under certain conditions. Hence, we added a few sentence in Section 3.1 (Line 139 to 141 in the revised manuscript) to be more inclusive in consideration of modeling uncertainty for offshore wind.

2) The appropriateness of comparison to lidar measurements should be commented on. Part of the reason that the WRF simulated values under-estimate the TKE (e.g. fig 7) may be down to the sensitivity of a pulsed lidar to measuring TI in unstable conditions. The authors can refer to the paper:

Sathe, A., Banta, R., Pauscher, L., Vogstad, K., Schlipf, D., Wylie, S., 2015. Estimating Turbulence Statistics and Parameters from Ground- and Nacelle-Based Lidar Measurements. IEA Wind Task 32 Expert Report. ISBN 978-87-93278-35-6. This report indicates that pulsed lidars can measure a value of turbulence which is significantly higher than a sonic at 80m above the ground under unstable conditions.

**Response:** We thank the reviewer for providing the insights into the potential uncertainties in TI modeling from the perspective of lidar turbulence measurement and atmospheric stability. We agree this report is a great reference for us and the readers. We have included a few sentences from Line 421 to 424 in Section 4.2 including the reference to this report.

3) The assumption that the vertical component of turbulence can be neglected could be validated from the lidar measurements. This would be especially pertinent under unstable conditions.

**Response:** We thank the reviewer for the suggestion. To address the comment in relative importance of wind variances, we conducted additional analysis of observational datasets collected at the MVCO ASIT and we used them to characterize the distribution of observed wind variances and summarized results as following:

Figure below shows the fraction of three components of wind variances ($\frac{\sigma_u^2}{\sigma_u^2+\sigma_v^2+\sigma_w^2}$, $\frac{\sigma_v^2}{\sigma_u^2+\sigma_v^2+\sigma_w^2}$, $\frac{\sigma_w^2}{\sigma_u^2+\sigma_v^2+\sigma_w^2}$) normalized by the total variance as a function of stability (using buoy-measured air temperature minus SST as its proxy). The results suggest overall, the fractions of horizontal wind variances ($\sigma_u^2$ and $\sigma_v^2$) are larger than the vertical wind variance ($\sigma_w^2$). In addition, there is no evident correlation between the fraction of $\sigma_w^2$ and stability. In most of the conditions, the fraction of $\sigma_w^2$ is no greater than 0.2. Only a few data points exceed 0.2 but they mostly occur during neutral conditions (Tair – SST is close to zero).

[Figure]

We further analyze what may cause those data having relatively large fraction of $\sigma_w^2$. By looking at the corresponding horizontal wind variances ($\sigma_u^2$ and $\sigma_v^2$) in function of fraction of $\sigma_w^2$ as in the figure below, we find that the relatively large fraction of $\sigma_w^2$ (larger than 0.4) is most likely due to concurrently small values in $\sigma_u^2$ and $\sigma_v^2$ (smaller than 1 m$^2$ s$^{-2}$).

[Figure]

A figure (Figure 2 in the revised manuscript) is added to illustrate the observed PDFs of three wind components of variances and their dependency on stability. Corresponding descriptions can be found in Section 3.2 from Line 202 to 261.

Although the paper is generally well written, there are a number of typos and instances of bade phrases that should be corrected by a thorough proof-reading.

**Response:** We thank the reviewer for the note. We've went through the manuscript and made appropriate corrections on the typos as well as the phrases were not accurately assigned.